# Influence of pre-existing multimorbidity on receiving a hip arthroplasty: cohort study of 28 025 elderly subjects from UK primary care

Rory Ferguson [1], Daniel Prieto-Alhambra [2], George Peat [3], Antonella Delmestri [1], Kelvin P Jordan [3], Vicky Y Strauss,[2] Jose Maria Valderas [4], Christine Walker,[5] Dahai Yu [3], Sion Glyn-Jones,[1] Alan Silman [1]

► http://dx.doi.org/10.1136/bmjopen-2020-046712

For numbered affiliations see end of article.

**Correspondence to**
Professor Alan Silman;
alan.silman@ndorms.ox.ac.uk

## ABSTRACT

The median age for total hip arthroplasty (THA) is over 70 years with the corollary that many individuals have multiple multimorbidities. Despite the predicted improvement in quality of life, THA might be denied even to those with low levels of multimorbidity.

**Objective** To evaluate how pre-existing levels of multimorbidity influence the likelihood and timing of THA.

**Setting** Longitudinal record linkage study of a UK sample linking their primary care to their secondary care records.

**Participants** A total of 28 025 patients were included, based on the recording of the diagnosis of hip osteoarthritis in a national primary care register, Clinical Practice Research Datalink. Data were extracted from the database on background health and morbidity status using five different constructs: Charlson Comorbidity Index, Electronic Frailty Index and counts of chronic diseases (from list of 17), prescribed medications and number of primary care visits prior to recording of osteoarthritis.

**Outcome measures** The record of having received a THA as recorded in the primary care record and the linked secondary care database: Hospital Episode Statistics.

**Results** 40% had THA: median follow 10 months (range 1–17 years). Increased multimorbidity was associated with a decreased likelihood of undergoing THA, irrespective of the method of assessing multimorbidity although the impact varied by approach.

**Conclusion** Markers of pre-existing ill health influence the decision for THA in the elderly with end-stage hip osteoarthritis, although these effects are modest for indices of multimorbidity other than eFI. There is evidence of this influence being present even in people with moderate decrements in their health, despite the balance of benefits to risk in these individuals being positive.

## INTRODUCTION

Total hip arthroplasty (THA) is a highly cost-effective procedure for hip osteoarthritis (OA).[1] The mean age at surgery in many Organisation for Economic Co-operation and Development (OECD) member countries is currently around 70 years,[2] and consequently,

**STRENGTHS AND LIMITATIONS OF THIS STUDY**

⇒ National sample of older patients recorded in primary care with newly reported osteoarthritis of the hip.
⇒ Multidimensional approach to assessing their concurrent morbidity and health status.
⇒ Linkage to national hospital records to assess receipt of surgery.
⇒ The challenge of using such routine data sources is to quantify the completeness and accuracy of diagnostic and treatment records.
⇒ Not able to assess impact on multimorbidity on referral as opposed to surgery.

many patients being considered for THA will have multiple comorbidities.[3] Our anecdotal experience, reinforced by the patients consulted prior to this research, was that pre-existing morbidities raise concerns about operative risk and hence deter surgery thereby limiting access to those who may otherwise benefit. Balancing risks and benefits of surgical procedures, for example, invasive cardiac procedures, has been well discussed for such potentially life-threatening disorders.[4] There is however very little evidence to guide healthcare professionals on the risks and benefits of THA in elderly patients with multimorbidity but who have severe limitation of quality of life.[5 6] In one report recently published from a tertiary care centre from the USA, background comorbidity was linked to both an increased risk of complications and a poorer outcome.[7] The concern, though, is that those with relatively modest background levels of multimorbidity may be excluded from surgery. Our objective was to determine, using a national primary care database, how levels of pre-existing

multimorbidity influenced the likelihood and timing of receiving THA.

## METHODS
### Design
In brief, we undertook a cohort study analysing routinely collected data gathered from the UK's Clinical Practice Research Datalink (CPRD) (ISAC approval number 17_024R).[8] This database of longitudinal primary care records was interrogated to identify elderly patients (aged 65 years and over) with a recorded diagnosis of hip OA. From these individuals, data from their CPRD record were used to extract information on their general health and disease status from which five approaches to scoring their multimorbidity were derived. Patients were then followed up to identify those recorded as receiving a THA. CPRD records were linked with records from the English Index of Multiple Deprivation (IMD) and the Office for National Statistics mortality dataset for data on secio-economic status and death.

### Subjects
These were 28 025 patients who had been diagnosed with hip OA according to electronic medical records in the CPRD GOLD database (see online supplemental appendix 1) between 1 January 1995 and 30 April 2014. The diagnosis of hip OA was based on being recorded with at least one from a list of relevant READ codes (online supplemental appendix 2).[9] A validation exercise was performed, in which clinical and radiographic data were obtained on a small subsample of 119 of these patients. Based on these detailed data, 88% were confirmed as having hip OA, which showed a sufficiently high level of diagnostic accuracy against accepted clinical criteria.[10]

### Multimorbidity markers
Information extracted from the longitudinal primary care records was used to score patients into different categories of five separate approaches to assessing their background morbidity and health at the time of the first recording of hip OA. The spectrum of measures aimed to cover: (1) specific risk scores accepted as predictive for postsurgical morbidity, (2) more generalised scores of health and function and (3) scores of healthcare utilisation. The specific scores used were:
1. Simple count of ever mention of the 17 chronic disorders included in the Quality and Outcomes Framework[11] (online supplemental appendix 3), a scheme to ensure maximal primary care compliance with recording chronic disorders.
2. Charlson Comorbidity Index (CCI) based on the cumulative burden of its components over the previous 5 years. The CCI is a well-accepted measure of perioperative and postoperative morbidity[12]
3. Electronic Frailty Index (eFI), a validated tool that aims to score frailty based on the number of different

'deficits' mentioned in the primary care record across a large number of health domains.[13]
4. Count of the number of different individual medications in the previous 12 months via CPRD GOLD prescription data.
5. Count of consultation events with primary care in the previous 12 months as a measure of health services burden.

### Follow-up
Patients were then followed up using their CPRD record to identify those where there was a record of receiving THA based on relevant codes in CPRD[14] by January 2015. As recording of THA may not be complete in CPRD, we also considered the alternative source ascertainment of THA based on linkage of CPRD records to the anonymised data from the national Hospital Episode Statistic (HES) dataset. HES linkage is only available for England; not all general practices in CPRD permitted linkage. We therefore conducted a sensitivity analysis (see further) using the HES data.

### Analysis
The analysis was directed towards assessing the relationship between the different categories of each of the multimorbidity scoring systems. Thus, we assessed both the cumulative incidence and the time to event of receiving a THA following the diagnosis of hip OA in the primary care record. There was a concern that in some cases the diagnostic label of hip OA may have been applied by the primary care physician once the decision to refer for surgery had been made. Thus, an arbitrary decision was made to exclude those who had surgery within 30 days from the diagnosis date as coded in the record. For each of the five approaches to scoring morbidity, patients were divided into four categories of increasing severity. The overall incidence of THA per 100 patient years was calculated for each category. The association between multimorbidity categories and the likelihood of THA was then investigated using Fine and Gray competing risk analysis models[15] with death being considered a competing event. The models are presented as morbidity-specific HRs with 95% CIs. The lowest multimorbidity category was defined as the referent category. Univariate and then multivariate models, adjusted for age, sex, region of the UK and calendar year of diagnosis, were conducted. We addressed the potential bias of the incompleteness of THA in the CPRD record. To assess this, we undertook a sensitivity analysis in the subset of practices with HES linkage using HES as a source of THA data.

### Patient and public involvement
This research was funded by the National Institute for Health Research Research for Patient Benefit scheme, which has an absolute requirement that there is patient and public involvement in all relevant stages of the research. The research question had been originally raised in a 'Priority Setting Partnership for Priorities

**Table 1** Demographic characteristics of cohort

| Variable | Category | N (28025) | % |
|---|---|---|---|
| Age (years) | 65–69 | 7200 | 25.7 |
| | 70–74 | 7229 | 25.8 |
| | 75–79 | 6421 | 22.9 |
| | 80–84 | 4233 | 15.1 |
| | 85–89 | 2161 | 7.7 |
| | 90+ | 781 | 2.8 |
| Gender | Male | 10662 | 38.0 |
| | Female | 17363 | 62.0 |
| BMI* | <18.5 | 257 | 1.2 |
| | 18.5–<25 | 6611 | 29.6 |
| | 25–<30 | 9569 | 42.8 |
| | 30–<35 | 4269 | 19.1 |
| | 35+ | 1638 | 7.3 |
| IMD quintiles | Least deprived | 4668 | 25.9 |
| | 2 | 4503 | 25.0 |
| | 3 | 3981 | 22.1 |
| | 4 | 3190 | 17.7 |
| | Most deprived | 1696 | 9.4 |

*No record of BMI in 5681.
†IMD data missing in 9987 as only available in England and on patients with linked records.
BMI, body mass index; IMD, Index of Multiple Deprivation.

**Table 2** Background morbidity status of cohort

| Morbidity scale | Category | N 28025 | % |
|---|---|---|---|
| Charlson Comorbidity Index | 0 | 18631 | 66.5 |
| | 1 | 3245 | 11.6 |
| | 2 | 3312 | 11.8 |
| | 3+ | 2837 | 10.1 |
| Count of chronic diseases | 0 | 9398 | 33.5 |
| | 1 | 8957 | 32.0 |
| | 2 | 5414 | 19.3 |
| | 3+ | 4256 | 15.2 |
| Count of medications prescribed | 0–4 | 10392 | 37.1 |
| | 5–7 | 7365 | 26.3 |
| | 8–12 | 6990 | 24.9 |
| | 13+ | 3278 | 11.7 |
| Count of contacts with primary care | 0–7 | 11945 | 42.6 |
| | 8–11 | 6378 | 22.8 |
| | 12–17 | 5212 | 18.6 |
| | 18+ | 4490 | 16.0 |
| Electronic Frailty Index | 0–4 | 19441 | 69.4 |
| | 5–8 | 7005 | 25.0 |
| | 9–12 | 1411 | 5.0 |
| | 13+ | 168 | 0.6 |

for Research in Hip and Knee Arthroplasty', and this was followed by a survey of members of the Keele PPI panel: Research Users Group (RUG). We then tested the suggested questions with the group and received very positive feedback. One of the RUG members (CW), herself a patient with direct experience of the target of the research, then became an active member of the research team, participated in all the meetings and advised on the design including the questions that should be asked during the analysis stage. As part of the dissemination phase, Keele PPI group organised a round table event attended by the lead authors where the results were discussed and guidance given on how those members of the public who were present wished to see disseminated.

## RESULTS

In all, 28025 patients aged over 65 years had been diagnosed with hip OA in their CPRD GOLD records between 1 January 1995 and 30 April 2014. Their mean age was 75 years and 62% were female. The details of their demographic characteristics are shown in table 1.

The distributions of morbidity data using the allocation of categories for the patients, at the time of first recording of their hip OA, are shown in table 2. The CCI scores showed that two-thirds had a score of 0, a similar proportion to those who had one or less other chronic disorder. The proportion of subjects in each category of increasing

score showed broadly similar trends for the three 'count' approaches of assessing morbidity. The frailty scores based on the eFI showed that only around 6% could be categorised as being 'moderately' or 'severely' frail.

On follow-up (figure 1), 11413 subjects had a THA at some stage. As discussed previously, we excluded 465 who had surgery within 30 days from the diagnosis date. Of the remainder, 10948 (39.7%) received a THA during the follow-up period with a median interval of just under 10 months, range 1 month to 17 years. The overall incidence of THA during the period of observation was 11.5/100 person years, that is, just over 10% per year. The THA rates by morbidity category are shown in table 3. In this crude analysis, there was variation in the relationship between the scores of the different measures of morbidity on whether or not a patient had surgery during follow-up. All the measures apart from the Charlson score showed a trend of decreasing surgery rates with increasing morbidity: the trend was steeper with eFI and only modest with the other measures. There was no trend of increasing surgery with the Charlson score.

The associations between each morbidity measure and time to surgery are shown in table 4. The cumulative incidence of surgery over time is also displayed graphically for each of the measures of morbidity in online supplemental figures 1-5. These results show a marked trend of decreasing likelihood of surgery with increasing levels of morbidity, however defined. The graphs show that the decreased likelihood of surgery is concentrated in

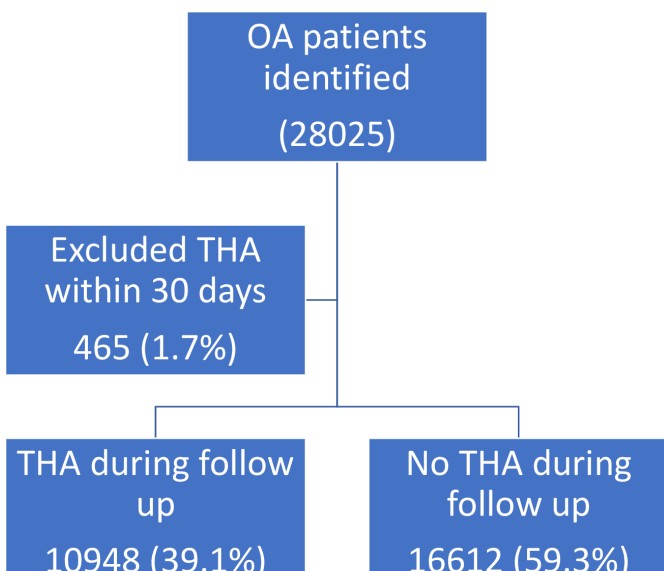

**Figure 1** Flow chart of participants (see attached file). OA, osteoarthritis; THA, total hip arthroplasty.

the earliest period of follow-up. Although the decreased occurrence of surgery was not unexpected, the reduction in surgery rates was observed even in those with the least increase in their morbidity status above 'baseline'. Thus, for example, those who were only 'mildly' frail or had a CCI of 1 were 23% and 16% less likely, respectively,

to have surgery. Adjustment for age and gender made little difference to the results suggesting that it was the morbidity and not age that explains the data.

## DISCUSSION

This paper provides a snapshot of the levels of background morbidity and health utilisation on large national cohort of elderly patients newly presenting to their general practitioner with hip OA. Not surprisingly given concerns about operative risk, the results showed that the greater the pre-existing morbidity, the longer the time to surgery. However, this influence on time to surgery is observed even in those with a low degree of multimorbidity. In this analysis, those who were just mildly frail had almost a quarter lower rate of surgery. Whether multimorbidity should influence decision for elective hip surgery has been debated given the relative lack of influence on long term mortality.[16] In a companion paper, the influence of multimorbidity on the outcomes of THA is examined, using the same UK primary care derived population, with the outcomes of improvement in pain, function and quality of life, and the rate of complications, revision surgery and death. the benefits of surgery in terms of quality of life. The results showed that benefits of surgery were substantial regardless of baseline morbidity, while the increase in

**Table 3** Crude incidence of total hip arthroplasty by morbidity measure

| Morbidity scale | Category | THA (N) | PYR | Incidence /100PYR | 95% CI |
|---|---|---|---|---|---|
| Charlson Comorbidity Index | 0 | 7878 | 64676 | 12.2 | 11.9 to 12.4 |
| | 1 | 1160 | 11729 | 9.9 | 9.3 to 10.5 |
| | 2 | 1115 | 10096 | 11.0 | 10.4 to 11.7 |
| | 3+ | 795 | 8628 | 9.2 | 8.6 to 9.9 |
| Count of chronic diseases | 0 | 4248 | 34744 | 12.2 | 11.9 to 12.6 |
| | 1 | 3631 | 30613 | 11.9 | 11.5 to 12.2 |
| | 2 | 1955 | 17482 | 11.2 | 10.7 to 11.7 |
| | 3+ | 1114 | 12289 | 9.1 | 8.5 to 9.6 |
| Count of medications prescribed | 0–4 | 4674 | 38337 | 12.2 | 11.8 to 12.5 |
| | 5–7 | 2911 | 24489 | 11.9 | 11.5 to 12.3 |
| | 8–12 | 2421 | 22576 | 10.7 | 10.3 to 11.2 |
| | 13+ | 942 | 9726 | 9.7 | 9.1 to 10.3 |
| Count of contacts with primary care | 0–7 | 4946 | 42662 | 11.6 | 11.3 to 11.9 |
| | 8–11 | 2559 | 21483 | 11.9 | 11.5 to 12.4 |
| | 12–17 | 1939 | 16902 | 11.5 | 11.0 to 12.0 |
| | 18+ | 1504 | 14080 | 10.7 | 10.1 to 11.2 |
| Electronic Frailty Index | 0–4 | 8373 | 69541 | 12.0 | 11.8 to 12.3 |
| | 5–8 | 2249 | 21349 | 10.5 | 10.1 to 11.0 |
| | 9–12 | 303 | 3839 | 7.9 | 7.0 to 8.8 |
| | 13+ | 23 | 399 | 5.8 | 3.4 to 8.1 |

PYR, per 100 patient years; THA, total hip arthroplasty.

**Table 4**  THA surgery rates by morbidity status

| Morbidity scale | Category | Total at risk | Competing events | THA events | Unadjusted HR (95% CI) | Adjusted HR* (95% CI) |
|---|---|---|---|---|---|---|
| Charlson Comorbidity Index | 0 | 18 303 | 2279 | 7878 | Ref | Ref |
| | 1 | 3202 | 630 | 1160 | 0.79 (0.75 to 0.84) | 0.84 (0.79 to 0.89) |
| | 2 | 3261 | 596 | 1115 | 0.78 (0.74 to 0.84) | 0.81 (0.76 to 0.87) |
| | 3+ | 2794 | 656 | 795 | 0.63 (0.58 to 0.68) | 0.65 (0.60 to 0.70) |
| Count of chronic diseases | 0 | 9228 | 1142 | 4248 | Ref | Ref |
| | 1 | 8802 | 1231 | 3631 | 0.90 (0.86 to 0.94) | 0.91 (0.87 to 0.95) |
| | 2 | 5339 | 881 | 1955 | 0.80 (0.76 to 0.84) | 0.81 (0.77 to 0.86) |
| | 3+ | 4191 | 907 | 1114 | 0.57 (0.53 to 0.61) | 0.59 (0.55 to 0.63) |
| Count of medications | 0–4 | 10 250 | 1206 | 4674 | Ref | Ref |
| | 5–7 | 7234 | 1111 | 2911 | 0.89 (0.85 to 0.93) | 0.92 (0.88 to 0.97) |
| | 8–12 | 6857 | 1169 | 2421 | 0.78 (0.74 to 0.82) | 0.81 (0.77 to 0.86) |
| | 13+ | 3219 | 675 | 942 | 0.64 (0.59 to 0.68) | 0.67 (0.62 to 0.72) |
| Count of contacts with primary care | 0–7 | 11 790 | 1570 | 4946 | Ref | Ref |
| | 8–11 | 6272 | 866 | 2559 | 1.00 (0.95 to 1.04) | 1.02 (0.97 to 1.07) |
| | 12–17 | 5097 | 825 | 1939 | 0.92 (0.88 to 0.97) | 0.95 (0.91 to 1.01) |
| | 18+ | 4401 | 900 | 1504 | 0.82 (0.77 to 0.87) | 0.85 (0.81 to 0.91) |
| Electronic Frailty Index | 0–4 | 19 102 | 2596 | 8373 | Ref | Ref |
| | 5–8 | 6900 | 1157 | 2249 | 0.75 (0.71 to 0.78) | 0.77 (0.74 to 0.81) |
| | 9–12 | 1391 | 354 | 303 | 0.48 (0.43 to 0.54) | 0.52 (0.47 to 0.59) |
| | 13+ | 167 | 54 | 23 | 0.31 (0.20 to 0.46) | 0.34 (0.22 to 0.51) |

*Age, gender, region and year of diagnosis.
THA, total hip arthroplasty.

risk of complication or death was modest, even for those with the highest levels of baseline morbidity.[17]

### Strengths and limitations

The population studied was derived from a national population database, CPRD GOLD, and provided a representative sample of patients in primary care with OA of the hip.[8] A primary care data set is also the only option for providing information on the levels of morbidity at the time of recording the hip diagnosis. These could then be analysed prospectively to evaluate the relationship between morbidity and the likelihood of surgery. Multimorbidity is a concept with no single measure that is perfect.[18] We used a broad range of measures, each identifying a different concept of multimorbidity that might have influenced surgical decision making.

There are a number of limitations that must be considered before interpreting the results further.

These data are based on the primary care recording of OA of the hip in this electronic dataset. There were likely to be errors both in terms of under-reporting of all cases and diagnostic inaccuracy in those who were included. We (DY, GP and KPJ) have previously reported on a retrospective analysis of the presence of codes for hip OA specifically, and OA in general, in patients who had a THA recorded in CPRD.[19] In that analysis, there was substantial under-reporting of hip OA when relying on a specific code of hip OA, although

the under-recording was lower when allowing the primary care to mention OA in any joint. The latter would have been too non-specific for the current analysis. By contrast (as mentioned previously), the level of diagnostic inaccuracy in those who were recorded as having hip OA, based on a detailed clinical review of a sample of cases, was low.[10] We also have no data on laterality so we had to assume that the side of the recorded OA was the same side as the surgery.

The major aim of the analysis was to study the time to surgery following the recording of the diagnosis of hip OA. Patients attending their primary care physician with hip pain or other relevant symptom may have only been recorded as having OA once a decision to refer for surgery had been made. Indeed, given the potential for under-reporting of OA of the hip, it is perhaps more likely that the OA label may have been made on those with more definite and severe OA. We also made the assumption that the side of diagnosis of hip OA was the side that was operated on.

These points illustrate the challenges in relying on administrative databases from primary care to identify a complete population of cases. Thus, our results on the relationship between multimorbidity and THA can only refer to those with a hip OA code.

We are unable to separate from our data set the contribution that baseline multimorbidity made between the decision by *primary care to refer for an orthopaedic opinion*

from that made in *secondary care as to whether to have surgery*. Although in practice such a separation may be hard to interpret as the referring primary care physician may influence both the patient and the orthopaedic surgeon in their decision. In theory, referral data are captured in UK primary care, but in practice, this was missing for most of the patients. We are unable to address from these data at what stage and what were the predominant influences that lead to a greater reluctance for surgery.

Any study such as this reflects the decisions made in relation to factors operating in the UK health system at that point in time. Views about the need for surgery and secular changes in resource availability might have influenced our results. We therefore adjusted for year of diagnosis and found no difference. However, it is interesting to see, for example, how the recent COVID-19 epidemic might influence surgical decision making currently. Indeed, the CCI is being modified to collect such information.[20]

CPRD provides access to the detailed primary care record that permitted the unique opportunity to derive the multimorbidity scores analysed in this study. Given the routine nature of the data gathering, there is scope for errors in the accuracy of the individual components, and the limitations of CPRD are well described.[8] Multimorbidity as assessed by our scores is not static. We examined the influence of the scores at the time of first recording of hip OA. Apart from the healthcare utilisation type scores (drug counts and primary care visits), the scores were based on accumulation of multimorbidities (over 5 years for CCI and ever for eFI and disease count) to the time of diagnosis. Given the median time between primary care recording and surgery was under a year, we suggest that these scores are unlikely to have changed to any important extent.

The crude incidence rates of surgery show a decreasing trend with increasing level of morbidity. These trends in rate of surgery were more pronounced when using a survival analysis. In addition, a separate question regarding those for whom the decision for surgery had been made is whether there was any influence of health pre-operative health status on delay to surgery. This analysis suggests that the interval between diagnosis and surgery was, if anything, slightly shorter in those with worse health (see online supplemental table 1). One explanation for this somewhat paradoxical result is that once the decision had been made to have surgery in the more 'at risk groups', then there was no reason to have any additional delay.

We used CPRD GOLD as our source of data on THA. THAs would be recorded in CPRD GOLD providing the information is fed back to patients' general practitioners by surgeons and the information is coded in their medical records.

In the UK, data on THAs funded by the National Health Service are also available from HES, which can be linked to CPRD. However, HES does not cover Scottish, Northern Ireland or Welsh patients, and also linkage to HES was not available for all practices; a link was only possible for just under 17 000 of our cohorts. We did undertake a sensitivity analysis (online supplemental table 2) using that subset. Although the total numbers were lower, the morbidity distributions were similar as, reassuringly, were the relationships with the likelihood of surgery in this subset to the whole dataset.

Finally, there are key confounders, which might explain the results. Thus, both BMI and smoking may be key contributors to the decision as to whether a patient with OA has surgery. CPRD GOLD records are incomplete in regard to these variables, and there is also concern of overadjustment. BMI might be the major predictor of the likelihood of surgery and should be examined. BMI data were available in 22 344 (79.8%) of the cohort but at varying times in relation to the record of OA. Given the lack of data of BMI at the time of OA diagnosis, it was not considered appropriate to adjust for this in the subset with a BMI record. As a check of the impact of excluding BMI as a confounder, we found only a very modest relationship between BMI category and surgery (data not shown) and did not alter the associations with morbidity in that subset.

Although we have demonstrated in a companion paper that coexisting morbidity did not constrain clinical benefit,[17] this does not capture all the reasons why surgery does not occur. Patients supported by their healthcare professionals may judge that their other health issues are so predominant to increase the reluctance for surgery despite the possible benefit. When we presented the results of this study to our patient partners, this was not their expressed view, but this requires further investigation.

In conclusion, modest multimorbidity does influence the timing of when patients undergo THA in the UK, despite evidence that the benefits in such groups outweigh the harms.

**Author affiliations**
[1]Nuffield Department of Orthopaedics, Rheumatology, and Musculoskeletal Sciences, University of Oxford, Oxford, UK
[2]Centre for Statistics in Medicine, Nuffield Department of Orthopaedics, Rheumatology, and Musculoskeletal Sciences, University of Oxford, Oxford, UK
[3]Primary Care Centre Versus Arthritis, School of Medicine, Keele University, Keele, UK
[4]Health Services and Policy Research Group, Medical School, University of Exeter, Exeter, UK
[5]Research User Group, Primary Care Centre Versus Arthritis, School of Medicine, Keele University, Keele, UK

**Acknowledgements** We would like to acknowledge Professor Andy Judge for advice on statistical analysis at the earliest stages of this project. We are also grateful to Dr Andrew Clegg and Stephen Pye for guidance in the use of the electronic frailty index. Dr John Griffiths and Professor Nigel Arden provided very helpful advice at the design stage on the challenges of anaesthetic assessment and on studying osteoarthritis from an epidemiological perspective. We would also like to acknowledge the contribution of the patient members of the Research Users Group from Keele in for their input, especially in the interpretation of the results.

**Contributors** The study was conceived by AS with SG-J and DP-A and detailed protocol produced by these authors with input from GP, KPJ, JMV and DY. AD was responsible for extracting, preparing and ensuring the quality of the datasets for analysis; analysis was undertaken by RF supported by VS, DP-A and AS, and CW provided patient input into the design, approach to analysis and interpretation of the data. All authors contributed to the discussion of the results. The manuscript was prepared by AS and RF and reviewed by all authors.

**Funding** This report is independent research funded by the National Institute for Health Research (Research for Patient Benefit programme, PB-PG-0815–20024). This study is based in part on data from the Clinical Practice Research Datalink obtained under licence from the UK Medicines and Healthcare products Regulatory

Agency. The data are provided by patients and collected by the NHS as part of their care and support.

**Disclaimer** The views expressed in this publication are those of the authors and not necessarily those of the National Institute for Health Research or the Department of Health and Social Care.

**Competing interests** None declared.

**Patient consent for publication** Not required.

**Provenance and peer review** Not commissioned; externally peer reviewed.

**Data availability statement** Data are available on reasonable request. The data that support the findings of this study are available from CPRD but restrictions apply to the availability of these data, which were used under license for the current study, and so are not publicly available. The datasets extracted from CPRD and used in the analysis during the current study are available from the corresponding author on reasonable request and consistent with the permission received from CPRD.

**ORCID iDs**
Rory Ferguson http://orcid.org/0000-0002-3256-6434
Daniel Prieto-Alhambra http://orcid.org/0000-0002-3950-6346
George Peat http://orcid.org/0000-0002-9008-0184
Antonella Delmestri http://orcid.org/0000-0003-0388-3403
Kelvin P Jordan http://orcid.org/0000-0003-4748-5335
Jose Maria Valderas http://orcid.org/0000-0002-9299-1555
Dahai Yu http://orcid.org/0000-0002-8449-7725
Alan Silman http://orcid.org/0000-0001-8426-8925

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
