## [Reviewer comments · BMJ Open]

ARTICLE DETAILS

TITLE (PROVISIONAL)	Influence of pre-existing multimorbidity on receiving a hip arthroplasty: Cohort study of 28,026 elderly subjects from UK primary care
AUTHORS	Ferguson, Rory; Prieto-Alhambra, Daniel; Peat, George; Delmestri, Antonella; Jordan, Kelvin; Strauss, Vicky; Valderas, Jose; Walker, Christine; Yu, Dahai; Glyn-Jones, Sion; Silman, Alan

VERSION 1 – REVIEW

REVIEWER	Kuperman, EF Department of Internal Medicine, University of Iowa, Carver College of Medicine
REVIEW RETURNED	30-Dec-2020

GENERAL COMMENTS	This study could be hypothesis-generating, especially in light of other work (including by this team, referenced within the paper) that hip surgery may be highly beneficial even in populations with multiple comorbidities. Unfortunately, there is a discrepancy between the crude rates and the hazard ratios that warrants further analysis before this paper can support its current conclusions. Major concerns: 1. The results are not clearly represented in the conclusions. For example, in the abstract, the authors report that "all approaches...were associated with a reducing likelihood of THA with increasing level of poor pre-operative health...even at low levels of comorbidity[.]" However, the observed relationships are not monotonic in Table 3 (where they are adjusted for exposure) and it is not clear to me why these relationships are significantly changed even in the unadjusted HRs in Table 4 unless there is significant variation in the follow up time over this period OR we are seeing delayed surgeries (as opposed to fewer surgeries) in patients with more comorbidities.2. Given that the follow-up period is highly variable and that it appears that the time to surgery is more impacted than the incidence of surgery, a graphical representation of incidence of surgery could be highly useful (e.g. "surgery-free survival") either in the main body of the paper or in the supplementary material. Alternate approaches (surgery from 1-6 months as an endpoint, for example) might also be illustrative. Minor concerns:
---

	1. The abstract is somewhat misleading as to the duration of observation. While the abstract lists follow up as "up to 17 years," the median follow-up was less than 10 months. 2. While the clinical data show that patients with comorbidities have a net benefit from hip arthroplasty, most perioperative risk calculators would demonstrate increased risk of infection, delay of return home, etc. Risk aversion may have significant impact on both patient and provider decision-making even if the expected net benefit is similar for these patients in the long-term, and reflection in the discussion seems appropriate.
--	--

REVIEWER	Cnudde, Peter Swedish Hip Arthroplasty Register
REVIEW RETURNED	04-Jan-2021

GENERAL COMMENTS	Thank you for the manuscript. The question raised about the influence of Preoperative multimorbidity on the timing and/or whether patient will receive a THR in the presence of those comorbidities is interesting, but should be considered multifactorial. There are however some questions with regards to inclusion criteria. There are some patients with confirmed radiological OA of the hip(s) that are functioning well and would not benefit from arthroplasty. There are patients with diagnosis of cancer where sometimes an elective arthroplasty would not be beneficial as the prognosis might be too compromised to be contemplated for surgery. Many patients with dementia and stroke (depending on the severity...) will be treated conservatively as risk/benefit analysis will favour a non-surgical approach. With regards to the validity of the study, I have some concerns:  1. THA can be offered for certain type of hip fracture. Are these included? Did you consider laterality? 2. Why not only including patients with available HES data? 3. Why excluding the patients with a THA quickly after the diagnosis? Likely to have received private treatment? 4. objective: likelihood of receiving a THA. Is that binary question Y/N or as stated in the conclusion to study the delay. This should really be clarified and worked out accordingly. I am a little worried that using these type of administrative databases might be inappropriate and incomplete to answer the research question. Most of the patients, currently, will following the referral be seen and assessed by an intermediate service and in principle will be offered conservative treatment initially. When conservative measures are inadequate (Inappropriate or exhausted), patient will be referred to the secondary care for consideration of surgery. Using preferably a shared-decision-making type of discussion patient will then be listed for surgery and reviewed by the pre-assessment team, where patient should be optimised and prepared for surgery (in the majority of cases). The current study does not analyse what happens to the patients along the pathway and is very high level.  5. the reference to COVID is in my view inappropriate, despite the timing. 6. I have some concerns on the quality/style of the references. Reference 2 (NJR annual report). reference 17? Reference 20? 7. The manuscript and abstract misses words and punctuation signs. 8. The influence of BMI on timing of surgery is currently a hotly debated issue and can influence the timing of surgery in some
--

	areas more than in other areas. The lack of BMI data and the timing of BMI collection is certainly a big limitation. 9. What is CPRD GOLD? PYR?
--	--

REVIEWER	Wilkinson, J. Mark Sheffield Teaching Hosp NHS Fdn Trust
REVIEW RETURNED	17-Jan-2021

GENERAL COMMENTS	This is a very interesting paper that seeks to explore the impact of co-morbidity on the rate of hip replacement in primary care patients with newly diagnosed hip OA, including 17 years of longitudinal follow up for incident hip replacement surgery. The findings add to existing knowledge by exploring what happens to the pool of OA patients in primary care in respect of their co-morbidities and subsequent selection for surgery. Major comments: The findings illustrate the selection process favours progression to surgery in patients with fewer comorbidities. However, the effect sizes and their impact on interpretation are somewhat overstated as they focus on the statistical impact rather than the actual hazard ratio change. The differences are for the majority of patients quite modest. One could thus argue that the provision of hip replacement in the UK remains remarkably consistent despite increasing number of co-morbidities, and that it is only really in the frailty score and highest other co-morbidity groups that provision consistently decreases. This will alter reader's perception of the conclusions, and is perhaps the more accurate interpretation. As an aside, where comorbidity influences THA rates, what is not clear is at what level(s) or by whom this selection operates: by the patient, primary care, or secondary care, or all 3. But this is not the function of this paper to address. Minor comments: 1) P3 Abstract Results: Although statistically true in places, the reporting of the results in the abstract does not sit comfortably with the actual data in Tables 3& 4 that show really only a fairly modest effect of most of the comorbidity scores on rates of THA until the highest groupings, and some noise between scores. The strongest consistent message is the frailty score. The authors could revise the results description in the text to better reflect the data findings. 2) P3 line 44, this first conclusion should be tempered by adding "...although these effects are modest for most indices of comorbidity with the exception of eFI". The next statement should say that the selection effects are for the most part small but associated with modest decrements in health... 3) P4 Strengths/limitations: "Large" is a relative term that depends upon perspective. In national terms this sample is actually small. Remove "large" from statement. 4) P5 line 10: give specific references for these stated anxieties about risks and impact on access to interventions. 5) P5 line 33: Context needed. The scope and coverage of CPRD GOLD over this timeframe should be clearly described. What percentage of UK practices and what percentage of UK patients?
---

	6) P5 line 48: Give the specific percentage number on the capture rate, rather than a subjective assessment of its quality. 7) P7 line 18: Sentence does not scan, reword. 8) P7 line 47: The basic demographics illustrate a difference between the CPRD data on OA diagnosis and national data on age at hip replacement, which is several years younger. As diagnosis of hip OA is not made 'post-hoc' after joint replacement surgery, the authors should discuss this discrepancy somewhere in the text
--	---

VERSION 1 – AUTHOR RESPONSE

2. Reviewer: 1

2.1. This study could be hypothesis-generating, especially in light of other work (including by this team, referenced within the paper) that hip surgery may be highly beneficial even in populations with multiple comorbidities. Unfortunately, there is a discrepancy between the crude rates and the hazard ratios that warrants further analysis before this paper can support its current conclusions. This is addressed in answer to 2.2 below (this is Reviewer 1's first major concern)

Major concerns:

2.2. The results are not clearly represented in the conclusions. For example, in the abstract, the authors report that "all approaches...were associated with a reducing likelihood of THA with increasing level of poor pre-operative health...even at low levels of comorbidity[.]" However, the observed relationships are not monotonic in Table 3 (where they are adjusted for exposure) and it is not clear to me why these relationships are significantly changed even in the unadjusted HRs in Table 4 unless there is significant variation in the follow up time over this period OR we are seeing delayed surgeries (as opposed to fewer surgeries) in patients with more comorbidities.

The reviewer raises an interesting point. In fairness there is a 'monotonic relationship' with all the measures of morbidity apart from the Charlson which we now emphasise (Page 7, lines 25-30) but the trend is more pronounced in the survival analysis. We revisit this point in the discussion (Page 9, lines 37 to 41) take up the helpful suggestion to look at time to surgery in those who had surgery (new Supplementary Table 1 Page 17). Interestingly that did not make a difference and the assumption is that the crude analysis did not allow for deaths during follow up

2.3. Given that the follow-up period is highly variable and that it appears that the time to surgery is more impacted than the incidence of surgery, a graphical representation of incidence of surgery could be highly useful (e.g. "surgery-free survival") either in the main body of the paper or in the supplementary material. Alternate approaches (surgery from 1-6 months as an endpoint, for example) might also be illustrative).

These is a useful suggestion and the curves are now included as supplementary material (Supplementary Figures 1-5) and discussed in the results (Page 7, lines 32-38)

Minor concerns:

2.4. The abstract is somewhat misleading as to the duration of observation. While the abstract lists follow up as "up to 17 years," the median follow-up was less than 10 months.

Fair point, abstract amended (Page 3, 26)

2.5. While the clinical data show that patients with comorbidities have a net benefit from hip arthroplasty, most perioperative risk calculators would demonstrate increased risk of infection, delay of return home, etc. Risk aversion may have significant impact on both patient and provider decision-

making even if the expected net benefit is similar for these patients in the long-term, and reflection in the discussion seems appropriate.

Fair point, though this of course is (reasonable) conjecture on which we have no data to address. We have added a paragraph to the discussion Page 10, lines 22-27)

3. Reviewer: 2

3.1. Thank you for the manuscript. The question raised about the influence of Preoperative multimorbidity on the timing and/or whether patient will receive a THR in the presence of those comorbidities is interesting, but should be considered multifactorial. There are however some questions with regards to inclusion criteria. There are some patients with confirmed radiological OA of the hip(s) that are functioning well and would not benefit from arthroplasty. There are patients with diagnosis of cancer where sometimes an elective arthroplasty would not be beneficial as the prognosis might be too compromised to be contemplated for surgery. Many patients with dementia and stroke (depending on the severity...) will be treated conservatively as risk/benefit analysis will favour a non-surgical approach.

We strongly agree with this point and have added to the discussion on this (page 10, lines 22-27). A key issue for us in initiating this study was the feedback from patient groups in particular and indeed orthopaedic surgeons that patients with more numerous multimorbidities may be inappropriately not having surgery. Indeed, the purpose of the companion paper is as Reviewer 3 commented these data are helpful given that pre-existing multimorbidity did not constrain the benefits from surgery in this population. What is needed (and is now the subject of our further research) is a formal qualitative study to examine this

3.2. With regards to the validity of the study, I have some concerns: THA can be offered for certain type of hip fracture. Are these included? Did you consider laterality?

As we stated in the paper our objective was only to consider the timing of surgery for patients with osteoarthritis. THA for hip fracture patients is not normally an elective procedure. We were unable to consider laterality insofar as the data are not available. We assumed that the side of the osteoarthritis was the side of the THA, but it is reasonable to mention this (page 8, lines 39-41)

3.3. Why not only including patients with available HES data?

As we had stated HES linkage was only available in the more recent years and did not cover the whole of the UK. We have done a subgroup analysis on that sub-group and have added this as a supplementary table (Supplementary Table 2, Page 18) and referred to this (page 10, lines 7-9)

3.4. Why excluding the patients with a THA quickly after the diagnosis? Likely to have received private treatment?

In the UK it is highly unlikely that an individual would receive surgery immediately after a primary care diagnosis of osteoarthritis even in the private sector. Far more likely, as we suggest, that there was a delay in the GP coding which was precipitated by the fact of surgery. As the reviewer raised this point, we have clarified this section (Page 6, lines 31-33)

3.5. Objective: likelihood of receiving a THA. Is that binary question Y/N or as stated in the conclusion to study the delay. This should really be clarified and worked out accordingly

Fair point, we have amended the relevant sections of the manuscript where this was not clear (Page 3, line 9; page 6, line 30; page 10, line 30)

3.6. I am a little worried that using these type of administrative databases might be inappropriate and incomplete to answer the research question.

Most of the patients, currently, will following the referral be seen and assessed by an intermediate service and in principle will be offered conservative treatment initially. When conservative measures are inadequate (Inappropriate or exhausted), patient will be referred to the secondary care for

consideration of surgery. Using preferably a shared-decision-making type of discussion patient will then be listed for surgery and reviewed by the pre-assessment team, where patient should be optimised and prepared for surgery (in the majority of cases). The current study does not analyse what happens to the patients along the pathway and is very high level.

We absolutely accept this point though only in part as there are so many different paths to surgery in the UK. More importantly our view, and the stakeholder opinions we sought, were that in general the intermediate path steps in the UK do not routinely consider detailed aspects of multimorbidity. These are anecdotal reports so probably not appropriate to include.

The challenge is that the only data we have to address the detailed evaluation of different approaches to assessing multimorbidity are administrative datasets. Ideally we wanted to collect data on both time to referral as well as time to surgery which could have answered this point more fully, but the quality and the completeness of that aspect are not available from the UK primary care data (disappointingly!). We have added to the discussion on this point (page 9, line 17-20)

3.7. The reference to COVID is in my view inappropriate, despite the timing.

CoVid has raised the whole issue of referral for elective surgery in the UK as is well described in the media. Waiting times have shot up. We don't feel this is an integral part of the paper but have left it for now as the other reviewers did not refer to this

3.8. I have some concerns on the quality/style of the references. Reference 2 (NJR annual report). reference 17? Reference 20

We will defer to editorial policy on this. Reference 2 has been updated as since submission there is a more recent report. Reference 17 is the companion paper, which we hope will be accepted in parallel with this paper and we believe the web address for Charlson as it is now used is the most helpful

3.9. The manuscript and abstract misses words and punctuation signs.

The revised manuscript and abstract have been carefully proof read so hopefully no further issues

3.10. The influence of BMI on timing of surgery is currently a hotly debated issue and can influence the timing of surgery in some areas more than in other areas. The lack of BMI data and the timing of BMI collection is certainly a big limitation

We don't disagree but given the broad range of approaches to assessing multimorbidity in this paper is a limitation, we would question whether obesity is a 'big' limitation. As an example, frailty as measured by eFI tends to be associated with being less underweight so the issue is a complex one.

As we now state even on the data we had BMI was unlikely to have importantly confounded the relationships across all the measures we assessed. The Reviewer's point is helpful as the issue should not be ignored despite its challenges. Our discussion is now enhance to reflect some of these (Page 10, lines 16-18)

3.11. What is CPRD GOLD? PYR?

The full names added earlier (Page 3, line 17, Page 6, line 36)

4. Reviewer: 3 Comments to the Author:

This is a very interesting paper that seeks to explore the impact of co-morbidity on the rate of hip replacement in primary care patients with newly diagnosed hip OA, including 17 years of longitudinal follow up for incident hip replacement surgery. The findings add to existing knowledge by exploring what happens to the pool of OA patients in primary care in respect of their co-morbidities and subsequent selection for surgery.

4.1. Major comments:

The findings illustrate the selection process favours progression to surgery in patients with fewer comorbidities. However, the effect sizes and their impact on interpretation are somewhat overstated as they focus on the statistical impact rather than the actual hazard ratio change. The differences are for the majority of patients quite modest. One could thus argue that the provision of hip replacement in the UK remains remarkably consistent despite increasing number of co-morbidities, and that it is only

really in the frailty score and highest other co-morbidity groups that provision consistently decreases. This will alter reader's perception of the conclusions, and is perhaps the more accurate interpretation. As an aside, where comorbidity influences THA rates, what is not clear is at what level(s) or by whom this selection operates: by the patient, primary care, or secondary care, or all 3. But this is not the function of this paper to address.

Fair point and we don't disagree completely, we have toned down our conclusion. We have amended the abstract along the lines suggested in 4.2. below (Page 3, lines 32-33) and the conclusion (Page 8, line 10)

The issue about pathway to surgery is as this reviewer states "not the function of this paper to address" – which we would have liked to have investigated but is similar to the point made in 3.6 above and addressed (Page 9, lines 17-20)

4.2. Minor comments:

4.2.1. P3 Abstract Results: Although statistically true in places, the reporting of the results in the abstract does not sit comfortably with the actual data in Tables 3& 4 that show really only a fairly modest effect of most of the comorbidity scores on rates of THA until the highest groupings, and some noise between scores. The strongest consistent message is the frailty score. The authors could revise the results description in the text to better reflect the data findings.

Reflects point 4.2 above and changed (Page 3, lines 26-28)

4.2.2. P3 line 44, this first conclusion should be tempered by adding "...although these effects are modest for most indices of comorbidity with the exception of eFI". The next statement should say that the selection effects are for the most part small but associated with modest decrements in health...

We don't have a problem with this rewording (Page 3, lines 32-33)

4.2.3. P4 Strengths/limitations: "Large" is a relative term that depends upon perspective. In national terms this sample is actually small. Remove "large" from statement.

Happy just to keep just the word 'national' sample (Page 4, line 2)

4.2.4. P5 line 10: give specific references for these stated anxieties about risks and impact on access to interventions.

This is one of those clinical opinions that are reported by both patients and health care professionals, and voiced in our consultations prior to this research. We would argue that the presence or absence of a research study in another population at a different time would not really add to this. We have added that this was our anecdotal experience (Page 5, lines 5-6)

4.2.5. P5 line 33: Context needed. The scope and coverage of CPRD GOLD over this timeframe should be clearly described. What percentage of UK practices and what percentage of UK patients? The details about CPRD GOLD are useful but we were concerned about the necessary level to provide in this paper. For the interested reader we now give a full description in a new Appendix 1.

4.2.6. P5 line 48: Give the specific percentage number on the capture rate, rather than a subjective assessment of its quality.

These items are now provided (Page 6, lines page 34-35)

4.2.7. P7 line 18: Sentence does not scan, reword.

This has now been done (new Page 8, line 2)

4.2.8. P7 line 47: The basic demographics illustrate a difference between the CPRD data on OA diagnosis and national data on age at hip replacement, which is several years younger. As diagnosis of hip OA is not made 'post-hoc' after joint replacement surgery, the authors should discuss this discrepancy somewhere in the text

This is probably not a discrepancy as this study focused specifically on those aged over 65, so inevitably the age distribution would be shifted towards an older population. To be fair to this reviewer, if this was not clear then we need to emphasise this aspect again, which we have done (Page 7, line 8)

VERSION 2 – REVIEW

REVIEWER	Kuperman, EF Department of Internal Medicine, University of Iowa, Carver College of Medicine
REVIEW RETURNED	01-Apr-2021

GENERAL COMMENTS	The authors have greatly strengthened their manuscript and I feel like all of my original concerns have been addressed. The additional data analysis lends additional robustness to their conclusions. One minor concern: Abstract, results subsection: The phrase "peri-operative risk" implies the risk of morbidity/mortality. I believe that the authors instead meant to indicate "the risk of a patient progressing to THA."
--

REVIEWER	Cnudde, Peter Swedish Hip Arthroplasty Register
REVIEW RETURNED	01-Apr-2021

GENERAL COMMENTS	Unfortunately I am remaining very sceptical about the quality of the data, the lack of laterality and the presentation within the manuscript. The effort to map the whole pathway from primary care to surgery needs to be applauded. Despite previous suggestions, there remain uncorrected concerns. Not sure whether the paper is relevant and brings new information that will improve the care and access. In the abstract: results do not reflect analysis: Even low levels of comorbidity were associated with some increased peri-operative risk. That was not even analysed in this study, nor was it the aim. Just looking at the HES data table: 39% of the most frail patients have a competing event (death during the study period). This is in contrast to 20% in the best group... Is that the reason for the selection? The timing and likelihood of surgery and its association with comorbidities are the aims of the study, are the conclusions representative of table 4? Means and Medians? To me, looking at the table, it looks like the least frail have to wait longer for their surgery 'on average' which contradicts the conclusion.
--

REVIEWER	Wilkinson, J. Mark Sheffield Teaching Hosp NHS Fdn Trust
REVIEW RETURNED	28-Mar-2021

VERSION 2 – AUTHOR RESPONSE

Reviewer 1

Use of term “peri-operative risk” (page 2, line12)

We agree this is open to confusion, we had used ‘risk’ in the epidemiological sense of likelihood of an event, but have changed as reviewer suggests “the risk of a patient progressing to THA” (our page 3, lines 29-30)

Reviewer 2

Although the Reviewer applauds – “the effort to map the whole pathway from primary care to surgery”. However, his criticisms are in the main non-specific and their overall negative feel is not supported by the remainder of his review. There are no substantive criticisms of the objective, the methods, the choice of the data bases, the approach to analysis or the nature of the results produced. This is not surprising given the robust peer review needed to obtain an NIHR grant, the robust scientific peer review to access the CPRD and of course the detailed reviews already obtained by BMJ Open.

In our response below we have therefore made changes to the ms where these are warranted but have provided some general comments to the reviewer’s others comments

Specific comments:

1. “Unfortunately, I am remaining very sceptical about the quality of the data, the lack of laterality and the presentation within the manuscript”.
 - a. We refute the comment about the quality of data which are totally unsubstantiated. With large numbers of publications CPRD is internationally recognised as providing a very rich data set of population morbidity in primary care. Inevitably national routine data of this type is not research quality but this is more than compensated by providing a unique opportunity to examine different constructs of morbidity. Although obvious to most readers we have added two sentences: (our page 9 lines 31-34: CPRD provides access to the detailed primary care record which permitted the unique opportunity to derive the comorbidity scores analysed in this study. Given the routine nature of the data gathering there is scope for errors in the accuracy of the individual components and the limitations of CPRD are well described (8).)
 - b. We believe the issue of “laterality” is tangential at best. For clarification the reviewer refers to the possible issue the date of diagnosis of hip OA and the date of surgery is subject to the error that there were different sides involved. That this could be a

serious source of error, compared to the other issues we have appropriately highlighted is small.. We have however added a sentence: (our page 9, lines 7-8 to ensure this possibility is aired.

“We also made the assumption that the side of diagnosis of hip OA was the side that was operated on.”

2. Not sure whether the paper is relevant and brings new information that will improve the care and access.

This comment is very strange as this reviewer is one of the few researchers in the field that has used registry data to examine the relationship between pre-operative health and hip surgery outcomes. In their Swedish data, the pre-operative health data related only to the Charlson and the closely related Elixhauser scores. We are surprised therefore that someone so involved would not find the much broader approach to assessing pre-operative morbidity of interest and relevance! We have not found any comparable study addressing the same issue, even with a much more superficial approach to examining pre-operative health, as in this study

3. In the abstract: results do not reflect analysis: Even low levels of comorbidity were associated with some increased peri-operative risk. That was not even analysed in this study, nor was it the aim.

We disagree with this comment but realise that in our attempt to keep the abstract concise, we had not clarified that our use of the term “low levels of comorbidity” referred to levels above the lowest. We have now reworded the relevant parts of the abstract (our page 3, lines 27-30, lines 33-34) .

In answering the comment about our study aim, we refer the Reviewer to the objective which “was to determine, using a national primary care database, how levels of pre-existing multimorbidity influence the likelihood and timing of receiving THA” (our page 6, lines 15-16) and the results for each measure spanned all strata of morbidity from the most to the least healthy.

4. Just looking at the HES data table: 39% of the most frail patients have a competing event (death during the study period). This is in contrast to 20% in the best group... Is that the reason for the selection?

It is no surprise that the most frail would have an increased risk of death and it is also no surprise that, because of their level of frailty, they would be less likely to have surgery. However, that group was only 167 persons and we would hesitate to make too much of that small group, especially as frailty was only one of our 5 markers of comorbidity. We have therefore not changed the ms in response

5. The timing and likelihood of surgery and its association with comorbidities are the aims of the study, are the conclusions representative of table 4? Means and Medians? To me, looking at the table, it looks like the least frail have to wait longer for their surgery 'on average' which contradicts the conclusion.

Our main analysis was appropriately a survival one as opposed to 'ever surgery'. The reviewer confuses time to surgery as a censoring event in the survival analysis and the delay to surgery in those who had surgery (this was the new analysis in Supplementary Table 1 we inserted at the requested of the reviewer)> We believe that this is less informative as by definition it excludes those who did not have surgery, which is why we have placed it as a Supplementary file. We had perhaps not clarified this distinction enough and have expanded on this (our page 11, lines 3-11)